# The Thermal Properties and Degradability of Chiral Polyester-Imides Based on Several l/d-Amino Acids

**DOI:** 10.3390/polym12092053

**Published:** 2020-09-09

**Authors:** Chen Qi, Wenke Yang, Fuyan He, Jinshui Yao

**Affiliations:** Shandong Provincial Key Laboratory of Processing & Testing Technology of Glass and Functional Ceramics, School of Materials Science & Engineering, Qilu University of Technology (Shandong Academy of Sciences), Jinan 250353, China; qichenqlu@163.com (C.Q.); wkyang@qlu.edu.cn (W.Y.); hefuyan555@163.com (F.H.)

**Keywords:** chiral amino acids, polyester-imides (PEIs), degradation, thermal property

## Abstract

Eight kinds of chiral diacid monomers were prepared with amino acids with different side groups or configurations. Polyester-imides (PEIs) were synthesized from these diacid monomers and diphenol monomers through polycondensation reaction, and the performances and properties were compared with the chiral polyamide-imides (PAIs) previously synthesized by our work group. Their thermal properties were analyzed by thermo gravimetric analysis (TGA) and dynamic thermomechanical analysis (DMA), and it was found that the glass transition temperature (Tg) of PEI was mainly affected by the volume of side groups. Their degradability was studied through buffer degradation experiments, and the changes in their water contact angle, molecular weight, structure and appearance during the degradation process were characterized by contact angle tester, gel permeation chromatography (GPC), Fourier transform infrared spectroscopy (FTIR), and scanning electron microscopy (SEM). With degradation, the hydrophilicity of PEI was improved, and when amino acids with larger side groups or D configuration were introduced into the backbone of PEI, the degradability decreased.

## 1. Introduction

In recent years, polymer materials have been widely used in various fields of our production and life, and have become an indispensable part of our lives. However, the shortcomings of polymer materials, which are difficult to degrade and can easily cause pollution, have caused great harm to the ecological environment [1,2,3,4,5,6,7,8]. Therefore, the development of degradable polymer materials has attracted more and more attention.

Polyimide is well known for its excellent heat resistance, chemical resistance, dielectric properties and mechanical properties [9,10,11], however, its high rigidity, difficulty in processing, and ease with which it can cause pollution, limit its development [12,13,14]. Polyester is the most concerned material in the field of degradation, such as polylactide (PLA), while its brittleness and poor heat resistance limit its development [15,16]. Polyester-imides (PEIs) obtained by incorporating imide bond and ester bond into the polymer concurrently can overcome their shortcomings, and the introduction of amino acids improves its biocompatibility [17,18,19].

The process of experiment is shown in Figure 1. The main purpose of this paper is to check the influence of the side group structure and main chain configuration on the properties of PEI. By controlling the types of amino acids incorporated into PEI, eight kinds of PEIs with different side groups or configurations were synthesized, and the influence of amino acids structure and configuration on the thermal performance and degradability of PEI was evaluated through thermal test and degradation experiment [20,21,22]. Through comparisons with the degradable polyimide (PI) previously studied by our group [20,21], we found that PEI was degraded more in a shorter period of time, illustrating that the introduction of an ester bond can improve the degradation property of PI.

## 2. Materials and Methods

### 2.1. Materials

l-alanine, d-alanine, l-2-aminobutyric acid, d-2-aminobutyric acid, l-Leucine, d-Leucine, l-phenylalanine and d-phenylalanine were provided by Aladdin Industrial Co (Shanghai, China). Dimethyl sulfoxide (DMSO), *N*,*N*-dimethylformamide (DMF), and pyridine (Py) were provided by Tianjin Guangfu Fine Chemical Research Institute (Tianjin, China). Pyromellitic dianhydride (PMDA), 4,4′-dihydroxydiphenyl ether, LiCl and diphenyl phosphoryl chloride (DPCP) were provided by Macklin Biochemical Co (Shanghai, China). Pyridine (Py) was further purified by vacuum distillation before using, other chemicals were used directly without further purification.

### 2.2. General Characterization

Fourier transform infrared (FTIR) spectra were generated using a Thermo Fisher Nicolet IS10 spectrophotometer (New Castle, DE, USA) in KBr. Proton nuclear magnetic resonance ^1^H NMR spectra were carried out on a Bruker AVANCE Ⅱ 400 MHz spectrometer (Swiss) in deuterated dimethyl sulfoxide (DMSO-d6) as solvent. The glass transition temperatures and thermomechanical properties of polymers were recorded by 850 dynamic thermomechanical analysis (DMA; TA instruments, New Castle, DE, USA) instrument at a heating rate of 5 °C/min. Thermogravimetric analysis (TGA) on polymers was conducted on a TGA/SDTA851 System (Setaram, Caluire-et-Cuire, France) under a nitrogen (N_2_) atmosphere at a flow rate of 10 °C/min. Specific rotations ($\left[\alpha \right]\begin{array}{c} 25\\ D \end{array}$) were determined with concentration of 0.1010 g dL^−1^ in DMF at 25 °C by a MCP-200 polarimeter (Anton Par, Austria). Melting points (mp) were analyzed by Melting Point Microscope SGW X-4 (China). Elemental analyses were recorded using Elementar Vario EL model element analyses equipment (Germany). The molecular weights (M_w_) of polymers were analyzed with gel permeation chromatography (GPC) and multi-angle laser light scattering (MALLS, Dawn Heleos), using a linear MZGel SD Plus GPC column set (two columns, 5-μm particles, 300 × 8 mm) with DMF as eluent at room temperature with a flow rate of 1 mL/min and a concentration of the polymer of ca. 1 mg/mL. The calibration was based on polystyrene standard. The water contact angles of polymers were characterized in SL200B type contact angle tester (Kenuo, Wuhan, China).

### 2.3. Synthesis of Chiral Diacid Monomers

Chiral diacid monomers (2a–2d′) were synthesized by the condensation of various amino acids and PMDA according to our previous works (see step 1 in Scheme 1) [21,22,23,24]. The yields were obtained by calculating the weight ratios of the materials before and after the reaction. Yields and physical properties of the diacid monomers are shown in Table 1. It can be seen that they have high yields and melting points, and the specific rotation and elemental analysis of them are in line with the expectation.

### 2.4. Synthesis of Chiral Polyester-Imides (PEI3a–PEI3d′)

Chiral PEIs (PEI3a–PEI3d′) were synthesized by direct polycondensation of diacid monomers (2a–2d′) with 4,4′-dihydroxydiphenyl ether under the reported method (see step 2 in Scheme 1) [25,26,27]. Yields and physical properties of PEIs are shown in Table 2. It can be seen that they have high yields and molecular weights, and the specific rotation and elemental analysis are in line with the expectation.

### 2.5. Preparation of PEI Films

An amount of 0.25 g of PEI powder (the size of the powder was about 150 μm) was added into 10 mL DMF under stirring until completely dissolved. Then the solution was put into a Teflon mold with a diameter of 5 cm. After that, it was put in an oven with a temperature of 50 °C and vacuum drying for 24 h, a piece of clear tawny flexible PEI film was obtained.

### 2.6. Degradation of PEI Films in Buffer

The several kinds of PEI films were cut to quarter sizes, and added into 15 mL Tris-HCl (pH = 7.4) buffer, before being transferred to a constant temperature incubator at 37.5 ± 0.5 °C. The films were removed from the buffer and washed with a large quantity of deionized water every week, and then placed into a vacuum drying box at 40 °C to dry [21,28].

## 3. Results and Discussion

### 3.1. Characterization on Polymers

The structures of diacid monomers and PEIs were determined by FTIR and ^1^H NMR. Figure 2 displays FTIR spectra of chiral PEI3a–PEI3d, and FTIR spectra of PEI3a′–PEI3d′ are shown in Appendix A. In the IR spectra, the stretching vibrations of C‒H bonds on aromatic and aliphatic groups appear at 3023 to 2850 cm^−1^. The specific absorption peaks at 1776 and 1724 cm^−1^ correspond to the asymmetric stretching vibration and symmetric stretching vibration of carbonyl (C=O) on the imide ring, while the stretching vibration of C=O on the ester bond appears at 1660 cm^−1^. The specific absorption peak of C‒C stretching vibration on benzene ring skeleton exists at 1492 cm^−1^. The typical C‒N stretching of aromatic-imide is observed at 1384 cm^−1^, and the absorption peak of C‒O‒C asymmetric vibration on aromatic nucleus appears at 1180 cm^−1^. In addition, the absorption peaks at 822 cm^−1^ and 724 cm^−1^ can be attributed to the out-of-plane bending vibration of the C‒H bond on the benzene ring and the C=O bending vibration of the five-membered imide ring, respectively.

Appendix A displays ^1^H NMR spectra of PEI3a-PEI3d′. The ^1^H NMR spectrum of PEI3a is shown in Figure 3. The resonance absorption peaks at 8.28 ppm and 7.05 ppm are provided by the benzene ring protons in PMDA and 4,4′-dihydroxydiphenyl ether, respectively. In addition, the hydrogen proton on the chiral carbon appears at 5.42 ppm, and the methyl proton of alanine appears at 1.62 ppm.

### 3.2. Thermal Properties

The thermal properties of PEI films were investigated through DMA and TGA. The transition temperature (Tg) values were received from DMA and the temperatures corresponding to weight loss of 5% (T_5_) and 10% (T_10_) of PEIs were acquired from TGA; these data are summarized in Table 3. The TGA curves of polymers are described in Figure 4. It can be seen that the TGA curves of all PEIs are roughly similar. As the volume of the side chain of amino acids increases, the rigidity of the polymers increases, and the T_5_ and T_10_ of PEI become larger and larger, which is called the “rigidity effect”.

Take PEI3a as an example, the T_5_ of it is above 410 °C, but after heating to 400 °C, the weight of it will drop rapidly, after heating to 800 °C, the loss of mass reaches more than 80%.

The TGA curves of PEI and polyamide-imide (PAI) are described in Figure 5. Comparing with PEI, the weight loss of PAI at 800 °C is about 55%, which is much higher than PEI, this is because at high temperatures, the oxygen atoms on the PEI ester bond react with carbon atoms to generate CO and CO_2_ gas, the gas volatilizes through the exhaust port in the TGA furnace, while the nitrogen atoms on the PAI amide bond react with the carbon atoms to generate solids, they remain in the crucible, so the weight loss of PAI is less than that of PEI. This is similar to the phenomenon when polyester fiber and polyamide fiber burn; when polyester fiber burns, a lot of black smoke is emitted, while when polyamide burns, it only melts. Although the residual qualities of these two polymers are quite different, they have already reacted to a large extent at high temperatures. The T_5_ and T_10_ of PAI are about 310 °C and 340 °C, respectively, which are much lower than those of PEI, so we think that the thermal stability of PEI is slightly better than that of PAI.

The thermomechanical properties and glass transition temperature values of PEIs were characterized by DMA. At 100 °C, the storage modulus of all PEIs are higher than 1500 MPa, which shows that they have high mechanical strength [29]. Figure 6 is the DMA curves of PEI3a, in the Tan δ curve, the transition between 140 and 170 °C corresponds to the β transition, and the peak at 225 °C corresponds to the α transition, that is, the glass transition. The appearance of the β transition is due to a relatively non-cooperative motion of the amorphous polymer segment, this phenomenon only occurs in amorphous polymers. As the degree of crystallinity and orientation increase, the β transition gradually disappears [29,30,31,32]. The crystallinity of PEI was investigated by XRD measurements. XRD curve of PEI3a is shown in Figure 7, all of the other PEIs have similar XRD curves. The XRD curve of PEI3a exhibits a broad band in the range 2θ = 15–30°, this indicates that PEI3a is completely amorphous. Thus, there is an obvious β transition in the Tan δ curve of PEI3a.

By comparing different structures of PEIs, it can be seen from the Table 3 that PEI3a has the highest Tg value, and the Tg from PEI3a to PEI3c decreases in turn. This is because with the increase in carbon chain in amino acids, the flexibility of polymer side chain will also increase, which leads to the decrease in Tg, this is called “internal plasticization” [33]. In general, the Tg of polymers increases with the increase in substituent rigidity, however, it can be seen that the Tg of PEI3d is the lowest. This is because the volume of the benzene ring on the side chain of PEI3d is much larger than that of the carbon chain on other polymers. The huge benzene ring leads to a larger distance between the PEIs, resulting in a smaller intermolecular force, which is called “distance effect” [34,35,36]. Therefore, the Tg value of PEI3d is the lowest. Furthermore, the reduction from PEI3a to PEI3c is very slow compared with the rapid decrease from PEI3c to PEI3d, which also shows that the Tg value of PEI is mainly affected by the distance effect. We can also find that the difference in Tg between PEIs of different chirality is not big, which shows that the structure of PEI is the decisive factor of its Tg.

### 3.3. Solubility

Table 4 displays the solubility of PEI3a–PEI3d′ in different solvents. The solubility performances of PEIs were studied quantitatively at a concentration of 0.5 g dL^−1^ and at room temperature in different solvents. The results demonstrate that PEIs are soluble in various solvents such as DMSO, DMF, THF, and sulfuric acid at room temperature, while in some other solvents, such as toluene, methylene chloride, methanol, ethanol and water, PEIs are insoluble.

In general, the solubility of PI in organic solvents is not high, the high solubility of PEI is due to the presence of branches on their structural units, which lead to a lower density of polymer molecular arrangements, thereby increasing the solubility of PEI [37].

Comparing with PAI, PEI has better solubility in different organic solvents, especially in THF. This is because the polarity of ester bond in PEI is less than that of amide bond in PAI, so PEI has better solubility in THF with smaller polarity. In addition, the molecular chain flexibility of PEI is higher, which makes it have better solubility.

### 3.4. Degradability Assay

Figure 8 displays the change of the water contact angles before and after the degradation of the PEI films. It can be seen that the water contact angles of the PEI films become significantly smaller after degradation, which is related to the increase in the hydrophilic groups exposed on the surface of the PEI films after degradation. With the degradation processing, the carbonyl groups in the main chains of the polymers were destroyed by microorganisms and hydrolysis, which exposed more hydroxyl and carboxyl groups on the surface of PEI films, resulting in smaller water contact angles and significantly improved hydrophilicity [38].

Figure 9 and Table 5 show the weight average molecular changes before and after degradation of PEI films. It can be seen that the weight average molecular remaining rates of PEI3a–PEI3d increase sequentially, which is related to the amino acid structure introduced by the polymer. Alkyl groups are hydrophobic groups, solvent molecules and microorganisms that play the role in degradation are hydrophilic substances, so the hydrophilicity of PEI will increase with the shortening of the carbon chain. The amino acid carbon chain of PEI3a is the shortest, only with one methyl group, so its degradation rate is the highest. The amino acid carbon chains of PEI3a–PEI3c increase sequentially, as a result, their degradation performances gradually decrease. In addition, the benzene ring on PEI3d is relatively stable, which is not easily destroyed by hydrolysis and microorganisms, so its degradation rate is the lowest.

Comparing PEI with different degrees of chirality, such as PEI3a and PEI3a′, we found that the weight average molecular remaining rate of PEI in the L configuration is less than that in the D configuration. This enantioselectivity stems from the spin polarization interaction. This mechanism for the intermolecular interaction of chiral molecules results in a higher degradation rate of the PEI in the L configuration [39]. Furthermore, by comparing the weight average molecular remaining rate of PEI with different chirality and structure, we found that the structure has a greater impact on degradation.

Figure 10 exhibits the IR spectra of PEI3a before and after degradation (the IR spectra of other polymers also exhibit close changes). It is not difficult to find that after 8 weeks of degradation, the asymmetric vibration of C=O bond at 1776 cm^−1^ and the symmetric vibration of C=O bond at 1724 cm^−1^ on the imide ring of PEI3a have almost no change, while the stretching vibration of C=O bond at 1660 cm^−1^ almost disappears, this demonstrates that in the degradation process of the PEI film, the ester bond is destroyed first, among which the carbonyl group in the ester bond is broken. Accompanied by the breakage of the carbonyl group, some small molecules are lost, which leads to the decrease in the weight average molecular weight of PEI. In addition, there are almost no change in other absorption peaks including the absorption peak of C‒C stretching vibration on benzene ring skeleton at 1492 cm^−1^ and the absorption peak of C‒O‒C asymmetric vibration on aromatic nucleus at 1180 cm^−1^, as well as the absorption peak of C‒H outside bending vibration on benzene ring at 822 cm^−1^. This is because ester bonds are easily broken by microorganisms and hydrolysis, while other groups in PEI, such as C-N bonds, are still not easy to break under the action of microorganisms and hydrolysis, which leads to the degradation of esters. The priority of bond degradation is much higher than that of other groups, so after degradation, the ester bond changes in PEI are the most obvious, while the changes in other groups are minimal.

It is well known that polymers based on amide bonds are not favored by microorganisms and hydrolysis [40,41]. Compared with PEI, there is not much difference between the amide bond of PAI and other groups in the priority of degradation, which leads to changes in many groups in the degradation process of PAI. The magnitude of the change is far less obvious than that of the ester bond in PEI. Therefore, we believe that PEI has better degradability than PAI.

Figure 11 displays the SEM images of the degradation process of PEI3a film. Before the degradation started, the surface of the PEI film was smooth, and after 4 weeks and 8 weeks of degradation, the surface of the film changed significantly. As the degradation time increased, more and more particles accumulated on the surface of the film. Moreover, due to the ester bonds on the polymer backbone chains were destroyed, the molecular weight of the polymer was decreased, which led the film became weaker and more easily to be broken. There are more hydrophilic groups on the PEI film, such as -COOH and -OH, which made the film more hydrophilic [37].

Figure 12 shows the SEM images of PEIs after degradation for 8 weeks. It can be seen that the surface of all PEI films has undergone significant changes, of which PEI3a has the most obvious change, which indicates that PEI3a has the best degradation performance. From PEI3a–PEI3d′, the number of particles on the surface gradually decreased, which indicated that their degradation performance also decreased sequentially.

## 4. Conclusions

Eight series of chiral PEIs (PEI3a–PEI3d′) based on eight diacid monomers (2a–2d′) have been successfully synthesized through direct polycondensation. By observing the structure of PEI and PAI, it is found that their difference is only in the ester bond and amide bond, while there are many similarities and differences in their properties. The degradation of PEI and PAI begins with the broken carbonyl group at the end of amino acid. The difference is that PEI has a higher degradation degree, while the degradation of PEI only occurs on the carbonyl group of the ester bond. This makes it possible to develop materials with specific degradation groups. By conducting thermal tests, we found that the T_5_ and T_10_ of PEI are higher than those of PAI, while PAI has a higher weight remaining rate than PEI at 800 °C. In terms of solubility, PEI has higher solubility in organic solvents. These phenomena can be attributed to the different properties of ester bonds and amide bonds. This provides ideas for the development of degradable materials with different functions.

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
