# Peer review of "The Thermal Properties and Degradability of Chiral Polyester-Imides Based on Several l/d-Amino Acids"

_polymers, 2020, doi:10.3390/polym12092053_

Round 1

Reviewer 1 Report

The paper entitled "The Thermal Properties and Degradability of Chiral Polyester-Imides Based on Several L/D-Amino Acids" by Chen Qi and co. presents the preparation of eight kinds of chiral diacid monomers and their characterisation.  

The paper is well organised and describes an interesting subject, but some major revision must be made:

1) in order to be more precise in describing the degradability of studied polymers, as follows:   

Pag. 7, row 143 statement "As the degree of crystallinity and orientation increase, this movement gradually disappears [27-30]." must be sustained by providing a cristallinity degree of all compounds, obtained from FTIR measuremets or, better, by performing XRD measurements, The cristallinity degree is nedded also for explaining the degratdation behaviour, by comparing the amount amorphous and the cristalline part of each studied sample. 

2) The" Conclusions" are a series of results already explained in the text. In this paragraph the authors must emphasize more the novelty and originality of the work.

Author Response

Response to Reviewer 1 Comments

To Reviewer 1:

Comments to the Author: “The paper entitled ‘The Thermal Properties and Degradability of Chiral Polyester-Imides Based on Several L/D-Amino Acids’ by Chen Qi and co. presents the preparation of eight kinds of chiral diacid monomers and their characterisation. The paper is well organised and describes an interesting subject, but some major revision must be made.”

Response: Thank you for affirmation of our research results, also thank you for your valuable comments. We have carefully considered the following comments and made necessary revisions accordingly by following your suggestions. We hope it could meet with your approval.

  1. Point 1: In order to be more precise in describing the degradability of studied polymers, as follows: Pag. 7, row 143 statement “As the degree of crystallinity and orientation increase, this movement gradually disappears [27-30].” must be sustained by providing a cristallinity degree of all compounds, obtained from FTIR measuremets or, better, by performing XRD measurements, the cristallinity degree is nedded also for explaining the degratdation behaviour, by comparing the amount amorphous and the cristalline part of each studied sample. 

Response: Thank you very much for the good and important advice. We agree with your comment. The crystallinity of PEI was investigated by XRD measurements. XRD curve of PEI3a is shown in the main document, all of the other PEIs have similar XRD curves. The XRD curve of PEI3a exhibits a broad band in the range 2θ=15-30°, this indicates that PEI3a is completely amorphous. Thus, there is a β transition in the Tan δ curve of PEI3a.

  1. Point 2: The “Conclusions” are a series of results already explained in the text. In this paragraph the authors must emphasize more the novelty and originality of the work.

Response: Thank you very much for the good and important advice. We agree with your comment. The “Conclusions” have been revised in accordance with your advice. We have changed it to “Eight series of chiral PEIs (PEI3a-PEI3d') based on eight diacid monomers (2a-2d') have been successfully synthesized through direct polycondensation. By observing the structure of PEI and PAI, it is found that their difference is only in the ester bond and amide bond, while there are many similarities and differences in their properties. The degradation of them begins with the broken of carbonyl group at the end of amino acid. The difference is that PEI has higher degradation degree, while The degradation of PEI only occurs on the carbonyl group of the ester bond. This makes it possible to develop materials with specific degradation groups. By conducting thermal tests, we found that the T5 and T10 of PEI are higher than those of PAI, while PAI has higher weight remaining rate than PEI at 800 °C. In terms of solubility, PEI has higher solubility in organic solvents. These phenomena can be attributed to the different properties of ester bond and amide bond. This provides ideas for the development of degradable materials with different functions.” We hope it could meet your approval after such modification.

Reviewer 2 Report

The problem considered at work is very important and undertaken for many years by scientists. The article deals with the problem of developing of degradable polymer materials. The problem is current and very significant in the light of the continuous increase in the use of polymers. There are many science works which try synthesize new materials. The authors place their hope in chiral polyester-imides based on several L/D-amino acids. Experimental tests, as i.e. thermal tests and chemical analysis (FTIR, GPC, HNMR) were performed.

The manuscript is interesting. However, there are several points that I would like to address:

  1. The introduction should be improved with more data in field of degradation time of this group of materials or properties in comparison to other degradable polymers. The literature data should be corrected and extended with data of the topic of the paper.
  2. Figure 1 - illegible descriptions in the diagram and reaction flask, figure SEM – should be written 0 week, not “weeks”.
  3. Line 40 – in the sentence: “The main purpose of this paper is to….” Instead of word “research” I propose: “check” or “test”.
  4. Line 62: instead of “America” I propose “USA”.
  5. Material and methods: how the yields were tested? Please describe method.
  6. Line 65 and 167 - the number of grams per unit volume is recorded differently (0.1010 g/100 mL or 0.5 g dL-1) - this should be standardized throughout the work.
  7. Table 1  - there is a lack of description of used abbreviations for physical properties, as: mp, α.
  8. Line 88: PEI powder- please characterize dimensions of particles of powder.
  9. Line 92: what means the description “same size” – please describe more precisely.
  10. What is the significance of the physical properties shown in the table 1 and in the table 2. Please discuss it in main text.
  11. Pages 6-8, 12: please justify the text.
  12. Between Figure and number the dot should be cancelled- please correct throughout the manuscript.
  13. Line 236: about what “colonies” there is discussion? Please describe it more precisely.
  14. Line 236: on what basis the sentence “ Moreover, the film became weaker and more easily to be broken, which indicates that the ester bonds on the polymer backbone chains were destroyed, resulting in increased rigidity” was formulated?
  15. English should be corrected in manuscript.
  16. Page 5: incorrect numeration of figures (line 98 and 110).
  17. Line 147: should be “Table_3”
  18. Line 148: instead of “largest” I propose “highest”.
  19. Conclusions, line 266: What means “Both of them have good thermal properties.”

Author Response

Response to Reviewer 2 Comments

To Reviewer 2:

The problem considered at work is very important and undertaken for many years by scientists. The article deals with the problem of developing of degradable polymer materials. The problem is current and very significant in the light of the continuous increase in the use of polymers. There are many science works which try synthesize new materials. The authors place their hope in chiral polyester-imides based on several L/D-amino acids. Experimental tests, as i.e. thermal tests and chemical analysis (FTIR, GPC, HNMR) were performed. The manuscript is interesting.

Response: Thank you for affirmation of our research results, also thank you for your valuable comments. We have carefully considered the following comments and made necessary revisions accordingly by following your suggestions. We hope it could meet with your approval.

  1. Point 1: The introduction should be improved with more data in field of degradation time of this group of materials or properties in comparison to other degradable polymers. The literature data should be corrected and extended with data of the topic of the paper.

Response: Thank you very much for this important advice. We agree with your comment. We have changed the second and third paragraph to “Polyimide is well known for its excellent heat resistance, chemical resistance, dielectric properties and mechanical properties [9-11], however, its high rigidity, difficulty in processing, and easy to cause pollution limit its development [12-14]. Polyester is the most concerned material in the field of degradation, such as polylactide (PLA), while its brittleness and poor heat resistance limit its development [15,16]. PEI obtained by incorporating imide bond and ester bond into the polymer concurrently can overcome their shortcomings, and the introduction of amino acids improves its biocompatibility [17-19].

The main purpose of this paper is to check the influence of the side group structure and main chain configuration on the properties of PEI. By controlling the types of amino acids incorporated into PEI, 8 kinds of PEIs with different side groups or configurations were synthesized, and the influence of amino acids structure and configuration on the thermal performance and degradability of PEI was evaluated through thermal test and degradation experiment [20-22]. Through comparing with the degradable PI previously studied by our group [20,21], we found that PEI was degraded more in shorter time, it illustrated that the introduction of ester bond can improve the degradation property of PI.” In addition, we have corrected and extended the references. We hope it could meet with your approval.

  1. Point 2: Figure 1 - illegible descriptions in the diagram and reaction flask, figure SEM – should be written 0 week, not “weeks”.

Response: Thank you very much for this comment. We have made a stupid mistake. We have made a correction and revised the word in Figure 1. We hope it could meet your approval after such modification.

  1. Point 3: Line 40 – in the sentence: “The main purpose of this paper is to….” Instead of word “research” I propose: “check” or “test”.

Response: Thank you very much for the good and important advice. We agree with your comment. We have replaced the term “research” with “check”. We hope it could meet your approval after such modification.

  1. Point 4: Line 62: instead of “America” I propose “USA”.

Response: Thank you very much for this important advice. We have replaced the term “America” with “USA”. We hope it could meet your approval after such modification.

  1. Point 5: Material and methods: how the yields were tested? Please describe method.

Response: Thank you for this comment. The yields were obtained by calculating the weight ratios of the materials before and after the reaction, and we have shown it in the main document. We hope it could meet your approval after such modification.

  1. Point 6: Line 65 and 167 - the number of grams per unit volume is recorded differently (0.1010 g/100 mL or 0.5 g dL-1) - this should be standardized throughout the work.

Response: Thank you for this comment. We have replaced the term “0.1010 g/100 mL” with “0.1010 g dL-1” We have made a correction in the revised main document. We hope it could meet your approval after such modification.

  1. Point 7: Table 1 - there is a lack of description of used abbreviations for physical properties, as: mp, α.

Response: Thank you very much for this important advice. The full name of mp is melting point, the meaning of α is specific rotation. We have given the full names of these abbreviations in the main text. We hope it could meet your approval after such modification.

  1. Point 8: Line 88: PEI powder- please characterize dimensions of particles of powder.

Response: Thank you very much for this good advice. According to our measurement, the size of PEI powder was about 150μm. We have shown it in the main document. We hope it could meet your approval after such modification.

  1. Point 9: Line 92: what means the description “same size” – please describe more precisely.

Response: Thank you for this comment. We have replaced the term “same size” with “quarter size”. We hope it could meet your approval after such modification.

  1. Point 10: What is the significance of the physical properties shown in the table 1 and in the table 2. Please discuss it in main text.

Response: Thank you for this comment. These physical properties can prove to a certain extent that the polymer we synthesized meets expectations, and we have discussed it in main text. We hope it could meet your approval after such modification.

  1. Point 11: Pages 6-8, 12: please justify the text.

Response: Thank you for this comment. We have justified the text in the main document. We hope it could meet your approval after such modification.

  1. Point 12: Between Figure and number the dot should be cancelled- please correct throughout the manuscript.

Response: Thank you for this comment. We have cancelled the dot between Figure and number. We hope it could meet your approval after such modification.

  1. Point 13: Line 236: about what “colonies” there is discussion? Please describe it more precisely.

Response: Thank you for this comment. We made a mistake in wording, and we have replaced the term “colonies and particles” with “particles accumulate”. We hope it could meet your approval after such modification.

  1. Point 14: Line 236: on what basis the sentence “Moreover, the film became weaker and more easily to be broken, which indicates that the ester bonds on the polymer backbone chains were destroyed, resulting in increased rigidity” was formulated?

Response: Thank you for this comment. We made a mistake in the expression of words. We changed this paragraph to “Moreover, due to the ester bonds on the polymer backbone chains were destroyed, the molecular weight of the polymer was decreased, which led the film became weaker and more easily to be broken.” We hope it could meet your approval after such modification.

  1. Point 15: English should be corrected in manuscript.

Response: Thank you for this comment. We have invited a more professional professor to revise English. We have corrected English in manuscript, for example, line 90, we have replaced the term “dry” with “drying”. Line 117, we have changed the word to “The Tg values were received from DMA and the temperatures corresponding to weight loss of 5% (T5) and 10% (T10) of PEIs were acquired from TGA are summarized in Table 3.” We hope it could meet your approval after such modification.

  1. Point 16: Page 5: incorrect numeration of figures (line 98 and 110).

Response: Thank you very much for your careful correction. We have made a stupid mistake. We have corrected the serial number of the figures in the main text. We hope it could meet your approval after such modification.

  1. Point17: Line 147: should be “Table 3”

Response: Thank you very much for your careful correction. We have changed it to Table 3. We hope it could meet your approval after such modification.

  1. Point 18: Line 148: instead of “largest” I propose “highest”.

Response: Thank you for this comment. We have replaced the term “largest” with “highest”. We hope it could meet your approval after such modification.

  1. Point 19: Conclusions, line 266: What means “Both of them have good thermal properties.”

Response: Thank you for this comment. We made a mistake in the expression of words. We changed this paragraph to “By conducting thermal tests, we found that all of PEIs have good thermal properties.” We hope it could meet your approval after such modification.

Round 2

Reviewer 1 Report

The paper was improved by introducing the responses to the reviewers comments

Reviewer 2 Report

All my comments were improved in revised version of manuscript. The  manuscript may be considered for publication.